# CAN ARTERIAL BLOOD PRESSURE PREDICT AGE? A CONVNET CLASSIFICATION TASK

## 1 INTRODUCTION

Blood pressure (BP) increases throughout life, and is controlled by several feedback mechanisms in mammals. Therefore, high resolution BP data may contain information related to the health and functionality of those systems, and the organism as a whole (Parati et al., 2023). Today, time series data like heart rate (Zhou et al., 2016) & (Fujiwara et al., 2016), 3D accelerometry (Le Goallec et al., 2023), EEG (Engemann et al., 2022), and ECG (Lima et al., 2021) are being successfully used to predict the onset/diagnosis of various diseases, and the occurrence of disease episodes. We believe BP data has the potential to achieve similar results.

Using surface-level measures like standard deviation and coefficient of variation (dispersion), approximate entropy and sample entropy (entropy), indices of detrended fluctuation analysis, and frequency domain analysis did not succeed in classifying age using BP data (Bakkar et al., 2021) & (Fares et al., 2022). According to (Martinez-Ríos et al., 2021), various shallow and deep learning methods have been extensively used on BP data to predict the class of established hypertension. As far as we know, there is little to no work done on the use of blood pressure data to estimate age among other physiological data, as well as diagnosing diseases other than hypertension.

## 2 METHODS

We collected beat-to-beat BP data from rats of different age groups, and attempted to classify their age (Young; 12-weeks vs. old; 24-weeks) using a 5-layer ConvNet. BP data were systematically down-sampled, and spectrograms were computed (**Figure 1A**) before being fed into the ConvNet. Saliency maps were generated to help understand the frequency and pixel utilization of 100 Hz BP data. Methods are discussed at length in **appendix A**.

## 3 RESULTS

Classification performance exceeded 90% at all BP sampling rates, and improved relatively as the sample rate increased, with the exception of the slight decrease in performance at 90 Hz sample rate (**Figure 1B**). In the 100 Hz BP data (**Figures 1C and 1D**), we observed that lower frequencies were utilized the most by the model, but the difference in utilization between frequencies was minute. Higher frequency components appear to contribute less but nonetheless could point to previously unknown spectro-temporal patterns that the model identifies for age classification.

### URM STATEMENT

The authors acknowledge that at least one key author of this work meets the URM criteria of ICLR 2023 Tiny Papers Track.

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

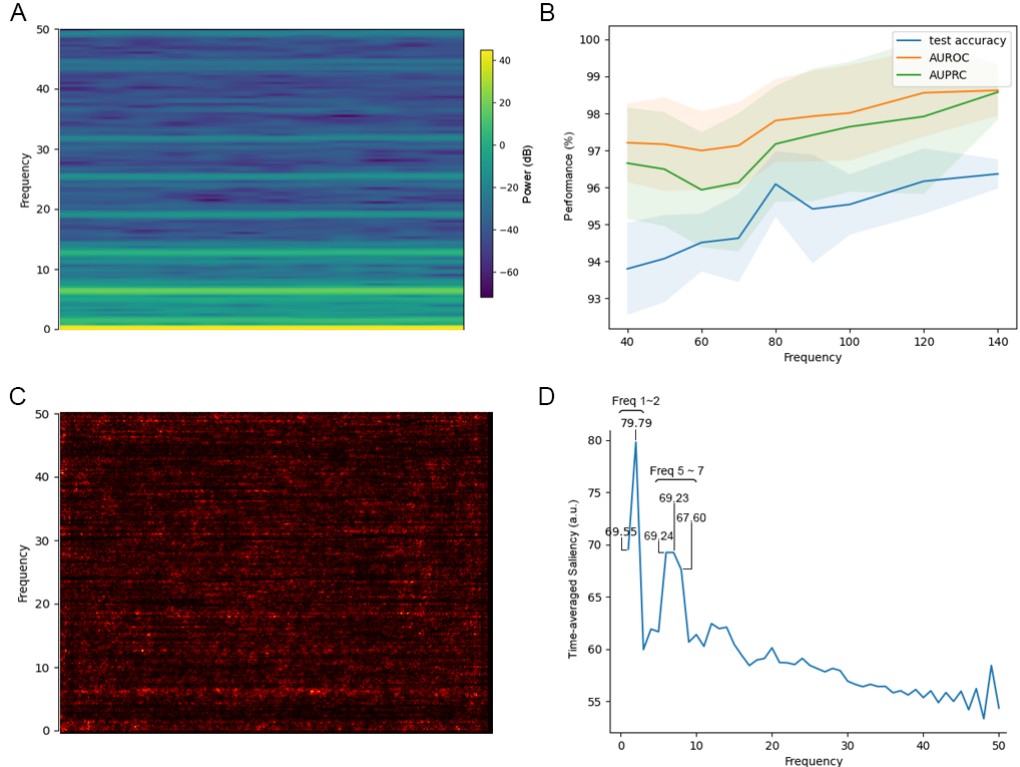

Figure 1: A: A sample spectrogram of a 12-week old rat. B: Model performance over BP sample rates 40-140 Hz. C: Saliency map of the same rat. D: Time-averaged saliency against frequency values in the 100 Hz sample rate BP data.

diction from M/EEG resting-state signals. *NeuroImage*, 262:119521, nov 2022. ISSN 1095-9572 (Electronic). doi: 10.1016/j.neuroimage.2022.119521.

Souha A. Fares, Nour Mounira Z. Bakkar, and Ahmed F. El-Yazbi. Predictive Capacity of Beat-to-Beat Blood Pressure Variability for Cardioautonomic and Vascular Dysfunction in Early Metabolic Challenge. *Frontiers in Pharmacology*, 13(June):1–15, 2022. ISSN 16639812. doi: 10.3389/fphar.2022.902582.

Koichi Fujiwara, Miho Miyajima, Toshitaka Yamakawa, Erika Abe, Yoko Suzuki, Yuriko Sawada, Manabu Kano, Taketoshi Maehara, Katsuya Ohta, Taeko Sasai-Sakuma, Tetsuo Sasano, Masato Matsuura, and Eisuke Matsushima. Epileptic Seizure Prediction Based on Multivariate Statistical Process Control of Heart Rate Variability Features. *IEEE transactions on bio-medical engineering*, 63(6):1321–1332, jun 2016. ISSN 1558-2531 (Electronic). doi: 10.1109/TBME.2015.2512276.

Alan Le Goallec, Sasha Collin, M'Hamed Jabri, Samuel Diai, Théo Vincent, and Chirag J Patel. Machine learning approaches to predict age from accelerometer records of physical activity at biobank scale. *PLOS Digital Health*, 2(1):e0000176, jan 2023. URL https://doi.org/10.1371/journal.pdig.0000176.

Emilly M Lima, Antônio H Ribeiro, Gabriela M M Paixão, Manoel Horta Ribeiro, Marcelo M Pinto-Filho, Paulo R Gomes, Derick M Oliveira, Ester C Sabino, Bruce B Duncan, Luana Giatti, Sandhi M Barreto, Wagner Jr Meira, Thomas B Schön, and Antonio Luiz P Ribeiro. Deep neural network-estimated electrocardiographic age as a mortality predictor. *Nature communications*, 12 (1):5117, aug 2021. ISSN 2041-1723 (Electronic). doi: 10.1038/s41467-021-25351-7.

Erick Martinez-Ríos, Luis Montesinos, Mariel Alfaro-Ponce, and Leandro Pecchia. A review of machine learning in hypertension detection and blood pressure estimation based on clinical and physiological data. *Biomedical Signal Processing and Control*, 68:102813, 2021. ISSN 1746-8094. doi: https://doi.org/10.1016/j.bspc.2021.102813. URL https://www.sciencedirect.com/science/article/pii/S1746809421004109.

Gianfranco Parati, Grzegorz Bilo, Anastasios Kollias, Martino Pengo, Juan Eugenio Ochoa, Paolo Castiglioni, George S Stergiou, Giuseppe Mancia, Kei Asayama, Roland Asmar, Alberto Avolio, Enrico G Caiani, Alejandro De La Sierra, Eamon Dolan, Andrea Grillo, Przemysław Guzik, Satoshi Hoshide, Geoffrey A Head, Yutaka Imai, Eeva Juhanoja, Thomas Kahan, Kazuomi Kario, Vasilios Kotsis, Reinhold Kreutz, Konstantinos G Kyriakoulis, Yan Li, Efstathios Manios, Anastasia S Mihailidou, Pietro Amedeo Modesti, Stefano Omboni, Paolo Palatini, Alexandre Persu, Athanasios D Protogerou, Francesca Saladini, Paolo Salvi, Pantelis Sarafidis, Camilla Torlasco, Franco Veglio, Charalambos Vlachopoulos, and Yuqing Zhang. Blood pressure variability: methodological aspects, clinical relevance and practical indications for management - a European Society of Hypertension position paper. *Journal of Hypertension*, 2023. ISSN 0263-6352. URL https://journals.lww.com/jhypertension/Fulltext/9900/Blood_pressure_variability__methodological.187.aspx.

Xin Zhou, Zhaolai Ma, Lingfu Zhang, Shuzhe Zhou, Jilian Wang, Bingyan Wang, and Wei Fu. Heart rate variability in the prediction of survival in patients with cancer: A systematic review and meta-analysis. *Journal of psychosomatic research*, 89:20–25, oct 2016. ISSN 1879-1360 (Electronic). doi: 10.1016/j.jpsychores.2016.08.004.

## A APPENDIX: METHODS

### A.1 ANIMAL SUBJECTS, AND DATA EXTRACTION

Here, we evaluate the capacity of a machine learning, convolutional neural network (CNN) in classifying arterial pressure signals collected from young vs. old rats. Four- to five-week-old Sprague Dawley (n = 139), male or female rats were fed a high-calorie or a normal diet for 12 or 24 weeks, representing young vs. old rats, respectively. A subset of female rats (n=12) underwent ovariectomy (removal of the ovaries); 6 of which were supplemented with estrogen, and another subset of male rats (n = 6) underwent myocardial infarction induction. At weeks 12 or 24, anesthetized rats were instrumented for invasive hemodynamics monitoring via a pressure transducer inserted through the carotid artery. Beat-to-beat arterial pressure signals of length 300 seconds were collected for each rat at a 1000 Hz sample rate. Age classification was carried on this heterogeneous population to simulate real-world data.

### A.2 MODEL ARCHITECTURE

Our model consisted of five convolutional layers, separated each by batch normalization, Re-LU activation, and max pooling. The outcome of the 5th convolutional layer had 40% of data dropped out and was delivered to a fully connected layer and then to a softmax function. The model utilized cross entropy for a loss function and Adam for an optimization function.

### A.3 DATA PREPROCESSING AND THE MODELING PROCESS

To understand the effect of sample rate on classification performance, we systematically downsampled the blood pressure data from 1000 Hz to 40-140 Hz (10 Hz increments from 40-100 Hz, 120 Hz and 140 Hz), and each sample rate was modeled independently for 10 times using different random seeds (producing a total of 90 runs). For each run, time series BP data were sliced into 16.67s bins, and Fourier-transformed to form a library of spectrograms for each age group. The model was run on each sample rate, and evaluated using classification accuracy, AUROC, and AUPRC. Models were trained for 30 epochs each, with a batch size of 25, 0.00005 learning rate, and no regularization. Eighty percent of the data were randomly allocated for training and 10% each for validation and testing.

## A.4 SALIENCY MAPS

We used the model trained on 100 Hz sample rate BP data to backpropagate the model accuracy score of the spectrograms, producing heat maps of the gradients of each pixel, which represent the degree to which they were utilized during the classification process. We then averaged all saliency maps into a single heat map to understand the patterns of pixel utilization across all spectrograms, and averaged all pixel columns across all saliency maps to investigate common patterns of frequency utilization.

