# OpenReview forum: "Can Arterial Blood Pressure Predict Age? A ConvNet Classification Task"
_ICLR.cc/2023/TinyPapers — Submitted to Tiny Papers @ ICLR 2023_

### Official Review · Reviewer_gxF3 · 2023-03-24

**Confidence:** 4

**Summary Of Contributions:**

The contribution of this work is to explore the potential of arterial blood pressure data to predict age through a convolutional neural network classification task. The authors found that classification performance exceeded 90% at all BP sampling rates, and improved relatively as the sample rate increased.

**Rating:**

High Potential (HP): a submission which meets the reviewing criteria and has potential to make an impact on the field

**Strengths And Weaknesses:**

Strengths:
1. The study uses a large sample size of rats, which increases the generalizability of the results.
2. The use of a convolutional neural network is a novel approach to predicting age using arterial blood pressure data.
3. The study provides insights into previously unknown spectro-temporal patterns that the model identifies for age classification.

Weaknesses:
1. The study only uses animal subjects, so it is unclear how well these findings will translate to humans.
2. The study does not provide information on how this method compares to other methods for predicting age.

**Suggested Changes:**

Potential Improvements:
1. Future studies could compare this method to other methods for predicting age, such as using heart rate or EEG data.
2. Future studies could also explore the potential applications of using arterial blood pressure data to predict other health outcomes beyond age, such as disease onset or progression.

---

### Official Review · Reviewer_tv7w · 2023-03-31

**Confidence:** 3

**Summary Of Contributions:**

Blood pressure data is taken from two groups of rats at either 12 or 24 weeks of age (young/old, respectively). A CNN is used to categorize these data into young or old.

**Rating:**

Needs Clarification (NC): a submission which does not meet the reviewing criteria and needs clarification for its described problem or solution

**Strengths And Weaknesses:**

Unfortunately, the presentation is very confusing and unclear to me. The paper is filled with medical jargon that is never defined. What is beat-to-beat blood pressure? No raw data is shown to give an intuition of the main question and setup to aid in understanding the details.

The importance of the stated goal of estimating age from blood pressure is not clear to me. Why can't we just ask a person how old they are or look at a form of identification? Also, if blood pressure increases throughout life, is it surprising that one can classify subjects' age into two categories of "young" and "old" if the age ranges are sufficiently far apart (12 weeks vs 24 weeks)? Why is it not possible to categorize the data using simpler models (not CNNs)?

What is shown in Figure 1C? What quantity is shown along the X axis of Fig. 1A and 1C? What is the unit? What data did you collect exactly? Why downsample the data? How would other parameter choices in the data preprocessing step affect the results?

**Suggested Changes:**

The paper needs to revised entirely. Unfortunately, I am too unclear about many parts to suggest specific points to improve, but the absolute most important point to consider in every point I am raising is to be clear to an audience that is not medically inclined.

It would be a great start to state the problem a lot more clearly for an audience with zero medical knowledge, why a complex model is needed to solve the problem, exactly what the approach is both in terms of data collection (what is being recorded and how?) and the prediction task (if X are the data and y are the labels, describe exactly how X and y are constructed), how the findings or the methods can form the basis of future work, and also perhaps state whether the data/code to reproduce the results are available.

---

### Meta-Review · Area_Chair_88cR · 2023-04-08

**Recommendation:** Invite to revise
**Confidence:** 4

**Metareview:**

This paper collects beat-to-beat BP data from rats of different age groups and uses a 5-layer ConvNet to classify the age. Overall, many details lack clarity, including the definition of the problem, how existing methods address the problem, and the motivation for the proposed approach (i.e., why 5-layer ConvNet is chosen).



**Summary:**

This work uses ConvNet to predict age from high-resolution blood pressure data. The two reviewers have opposite ratings. After reading the paper, the AC thinks that this paper would need efforts to improve the clarity.

**Reason For Not Giving A Higher Recommendation:**

The presentation needs significant improvements.

**Reason For Not Giving A Lower Recommendation:**

N/A

---

### Decision · Program_Chairs · 2023-04-08

No revision received; not invited to archive